# Learning Collision Situation to Convolutional Neural Network Using Collision Grid Map Based on Probability Scheme

**Jun Hyeong Jo and Chang-bae Moon *** 

School of Mechanical Engineering, Chonnam National University, 77 Yongbong-ro, Buk-gu, Gwangju 61186, Korea; 187127@jnu.ac.kr
* Correspondence: cbmoon@jnu.ac.kr; Tel.: +82-62-530-1664

**Abstract:** In this paper, a Collision Grid Map (CGM) is proposed by using 3d point cloud data to predict the collision between the cattle and the end effector of the manipulator in the barn environment. The Generated Collision Grid Map using x-y plane and depth z data in 3D point cloud data is applied to a Convolutional Neural Network to predict a collision situation. There is an invariant of the permutation problem, which is not efficiently learned in occurring matter of different orders when 3d point cloud data is applied to Convolutional Neural Network. The Collision Grid Map is generated by point cloud data based on the probability method. The Collision Grid Map scheme is composed of a 2-channel. The first channel is constructed by location data in the x-y plane. The second channel is composed of depth data in the z-direction. 3D point cloud is measured in a barn environment and created a Collision Grid Map. Then the generated Collision Grid Map is applied to the Convolutional Neural Network to predict the collision with cattle. The experimental results show that the proposed scheme is reliable and robust in a barn environment.

**Keywords:** 3d point cloud; classification; Convolutional Neural Network

---

## 1. Introduction

Many studies [1–4] are proposed where the robot is suitable for feeding and monitoring livestock in the barn environment. When a mobile robot feeds the cattle, there is a problem with the cow sticking their head out of the fence for eating and colliding with the end effector of the manipulator. In order to solve the collision problem, the K-Nearest Neighbor method [5] was used to cluster point cloud data to detect obstacles. The depth space method [6,7] was defined to find the obstacle in depth space and cartesian space. Obstacles were constructed with octree structures [8] to find a safe space. The ANN method [9,10] was modeled as an adaptive neural network for predicting penetration rate and rock tensile strength.

Recently, a Convolutional Neural Network (CNN) algorithm [11–16] using image data for object classification was proposed. In the proposed method [11], the neural network shows the feature of image data using gradient descent for optimizing. As one of the deep neural network models, Alex Net [12] is a network that is deeply constructed and extracts not only simple features but also high-level features to enhance the performance of the object classification. In the VGG Net [13], it uses a filter size that is smaller than Alex Net and an increased depth layer to enhance the performance. When deeper networks are optimized, there is a degradation problem that the neural network is not efficiently learned by training data. ResNet [14] addresses the degradation problem used for the residual learning method. A pheromone convolutional method [15] is used as discrete gaussian distribution to generate the efficient network path in a topological network. In PSI-CNN [16], it extracts

an untrained feature map from various image resolutions. The untrained feature map is used to fuse the original feature map. PSI-CNN shows stable performance when tested on low-resolution images acquired from CCTV cameras.

Unlike images that are regularized pixel, the 3D point cloud is unordered data. These characteristics occur as invariant permutation problems when point cloud data is applied to the Convolutional Neural Network. Some references [17–20] were proposed for the Convolutional Neural Network (CNN) using a 3d point cloud. In PointNet [21], an invariant problem is solved by a symmetric function formulated multi-layer perceptron (MLP) and a max pooling function. As the end-to-end deep neural network, PointNet is directly learned by point cloud, which segmented the features of 3d objects and demonstrated high performance in classifications. PointNet++ [22] is able to learn hierarchical features with increasing scales of contexts. The network of PointNet++ deals with non-uniform densities by using density adaptive strategy. In Reference [23], sparsity of point cloud was utilized by using sliding window detection with linear classifiers and a feature-centric voting method. Vote3deep [24] is able to efficiently learn a 3d feature by a sparse convolutional layer, which is based on the voting scheme. VoxNet [25] is learned by three types of the occupancy grid maps [26,27]. Learned network shows robust performance by a grid map, which is modeled by binary, hit, and density grid maps used for 3d point cloud from lidar or the RGBD camera. As the end-to-end learning network, proposed VoxelNet [28] extracts the 3D bounding box using 4-dimensional tensor and RGB data. It is based on a Region Proposal Network, a Convolutional Middle Layer, a Feature Learning Network, and a Voxel Feature Encoding (VFE) scheme.

In this paper, the Collision Grid Map (CGM) method is proposed by using 3D data to predict collision between the cow and the end effector of the manipulator in a barn environment. In a conventional method [25], the network is learned by three types of voxel mentioned above. However, in this paper, the Collision Grid Map is used and composed of a two-channel without any voxel. Each cell in the first channel is computed by distribution of location data in the x-y plane. The second channel consists of the average of depth z data. This proposed method is not a transformed locational feature of the point cloud in the x-y plane data. The neural network is learned about location features and safety distance without an invariant permutation problem. The Generated Collision Grid Map is applied to a Convolutional Neural Network for prediction of collision. In addition, learned network classifies the collision situation in the barn environment.

The remaining sections of this paper are organized as follows. In Section 2, the Collision Grid Map scheme and Convolutional Neural Network algorithm using the Collision Grid Map are explained. The experimental setup and results are described in Section 3. The concluding remarks are presented in Section 4.

## 2. Collision Grid Map Scheme and Convolutional Neural Network

### 2.1. Collision Grid Map Scheme

There is an invariant of the permutation problem, which is not efficiently learned in an occurring matter of a different order, as 3D point cloud data is applied to a Convolutional Neural Network. The proposed method [25] modelled voxel of the binary occupancy grid, the density grid, and the hit grid. Then the network is learned by voxels to solve an invariant of the permutation problem. However, the Collision Grid Map scheme with a two-channel is proposed to predict collision.

Equation (1) shows measurement data **s** and grid map **M**. Grid map **M** is constructed of the two-channel and $m_{1,i}, m_{2,i}$ is each of the grid channel.

$$\mathbf{s} = \{s_i\} = \{x_i, y_i, z_i\}, \ \mathbf{M} = \{m_{1,i}, m_{2,i}\} \tag{1}$$

The Posterior of the Collision Grid Map is obtained as shown in Equation (2). In order to calculate the posterior efficiently, log odds notation can be defined, as shown in Equation (3).

$$p(m|\mathbf{s}) = \prod_i p(m_i|\mathbf{s}) \tag{2}$$

$$l(m_i|\mathbf{s}) = \log\left(\frac{p(m_i|\mathbf{s})}{1 - p(m_i|\mathbf{s})}\right) \tag{3}$$

Equation (4) is shown as applying log odds notation to Equation (2). Log odds notation is calculated using present point cloud data and previous point cloud data. When an initial grid map is free state, the initial Collision Grid Map can be omitted.

$$l(m_i|s_{1:n}) = l(m_i|s_n) + l(m_i|s_{1:n-1}) - l(m_0) \tag{4}$$

Equations (5) and (6) show $l(m_i|s_n)$ that is modeled to predict the collision situation.

$$l(m_{1,i}|s_n) = G(x_n, y_n) \tag{5}$$

$$l(m_{2,i}|s_n) = \frac{\sum z_{1:n}}{m_{1,i}} \tag{6}$$

The first channel of the Collision Grid Map uses the discretization function **G** to grid the x and y data in the point cloud. When discretization data corresponds to the grid cell, grid cell values are added up in counts one by one such as in Equation (4). The second channel is consisted of the average of depth z data. The value of the cell in the first channel is constructed from x-y data based on the probability method. The value of the cell in the second channel is shown as a safe distance between the end effector and the cow.

The example of generating the first channel in the Collision Grid Map and Collision Grid Map is shown in Figure 1. When the Collision Grid Map is composed of a two-channel, an invariant of the permutation problem that occurs to apply point cloud to a Convolutional Neural Network is solved and raw point cloud data size becomes reduced. By modeling the Collision Grid Map, networks are able to learn not only distribution of the x-y plane data but also collision safety distance that is depth z data.

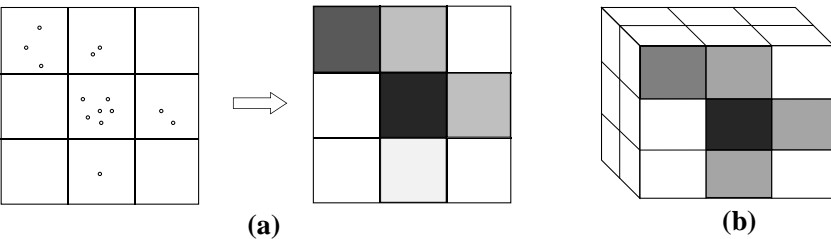

            **(a)**                                                    **(b)**

**Figure 1.** (**a**) The example of generating the first channel in a Collision Grid Map using raw point cloud and (**b**) an overall Collision Grid Map.

### 2.2. Convolutional Neural Network Algorithm Using the Collision Grid Map

The Convolutional Neural Network algorithm can be shown robustly in the office and road environment. However, there is required big data due to the different appearance. In an automatic farm system, typically fences and livestock have same appearance. Unlike the office and road environment, networks can be learned with less data that is defined as a collision situation. In order to predict the collision situation, the non-collision situation is defined when there is a fence only and the collision situation is defined when cows stick their head out of the fence. In these defined situations, performance of segmentation is guaranteed even without deep networks due to normalized data of the appearance.

The convolutional layer is learned features of local data in a Convolutional Neural Network algorithm. The proposed Collision Grid Map transforms 3D point cloud data to grid data without destroying spatial characteristics of point cloud data. When networks are learned, a generated Collision Grid Map can be effectively applied to the Convolutional Neural Network to predict the collision situation.

When weights of convolutional layers are w in N by N size, Equation (7) is shown as the learned convolutional layer for a generated Collision Grid Map. $c$ is the convolutional layer value and $m^*$ means the generated Collision Grid Map.

$$c_{ij} = \sum_{a=0}^{N-1} \sum_{b=0}^{N-1} \left( w_{ab} m^*_{(i+a)(j+b)} \right) \tag{7}$$

Figure 2 applies the Collision Grid Map to Convolutional Neural Network. The network is a learned spatial feature about collision and a non-collision situation in the convolutional layer. In order to reduce computation, data size is reduced in the pooling layer. In a practical barn environment, measured data about the defined situation is filtered by the learned network to detect the local feature of the x-y plane and safety distance. Then, the SoftMax function transforms the filtered data to a probabilistic value. A high probability score is selected as an appropriate situation. When the Collision Grid Map is applied to a Convolutional Neural Network, there are some advantages that the network is learned not only in local features in the x-y plane data but also on a safety distance in the z direction to distinguish between the collision and non-collision situation.

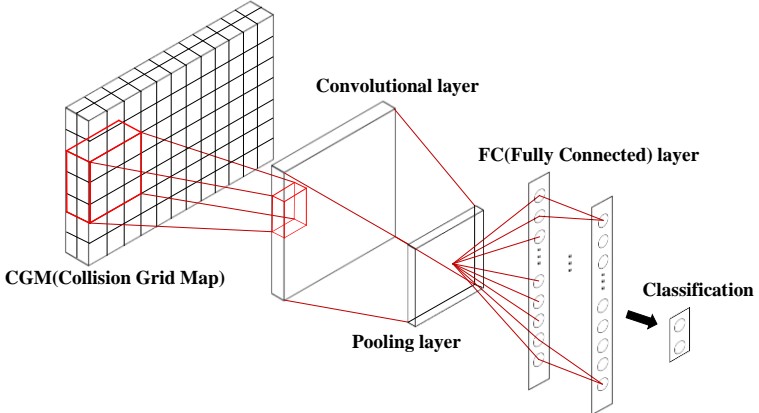

**Figure 2.** Applying the collision grid map to the convolutional neural network.

## 3. Experimental Results

### 3.1. Experimental Setup

3D point cloud data is measured in the barn environment using D435 from intel as shown in Figure 3a. D435 has a maximum resolution 1920 × 1080 and a maximum frame rate of 30 FPS. It can be measured up to 10 m and 90° about the depth diagonal field of view. Our robot has intel core i5 4300 U processor and 8 GB RAM. This experiment has been performed using the robot equipped with a D435 shown in Figure 3b. When networks are learned, the CPU Intel core i9-9900 K and GPU Nvidia GeForce RTX 2080 Ti are used.

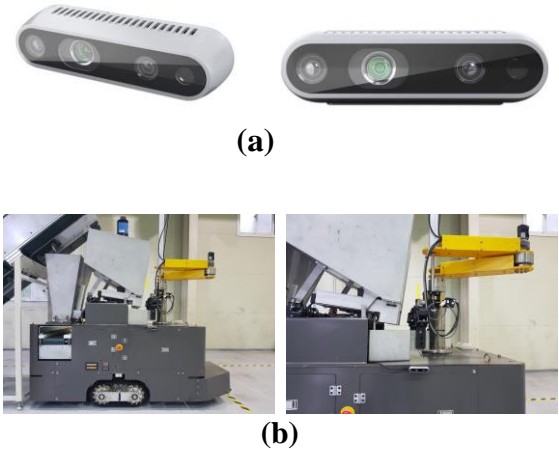

**(a)**

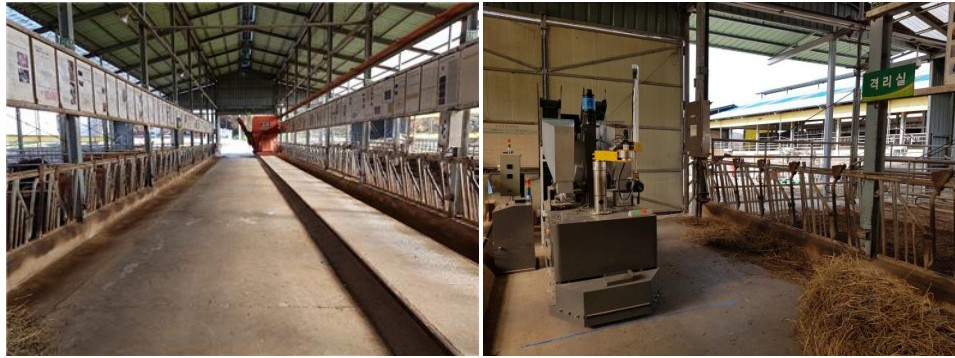

**(b)**

**Figure 3.** (**a**) D435 form Intel. (**b**) Robot equipped with a D435.

The experimented barn is 54.9 m in width and 6.05 m in height. Each section is 4 m wide and consists of a total of 10 sections. We experimented by setting the robot in a barn environment, as shown in Figure 4.

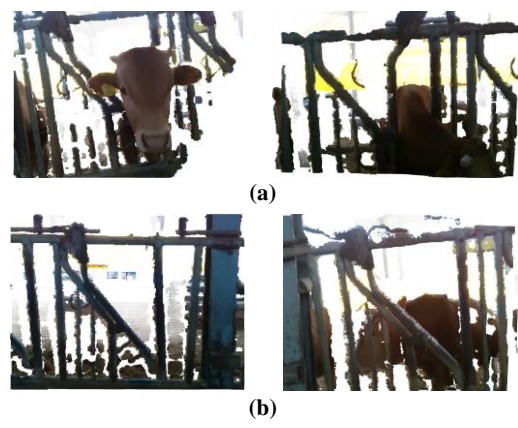

**Figure 4.** Appearance of the experimented barn environment.

The data in the case of non-collision situation, when there is a fence only, has been measured using 2000 data. Similarly, the data in the case of the collision situation, when cows stick their head out of the fence, has been measured using 2000 data. The data is divided into 1800 training data and 200 test data each. Figure 5a shows the example of 3D point cloud data about the collision situation. Furthermore, Figure 5b shows the example of 3D point cloud data about the non-collision situation.

**(a)**

**(b)**

**Figure 5.** (**a**) Example of 3D point cloud data about the collision situation. (**b**) Example of 3D point cloud data about the non-collision situation.

### 3.2. Result of the Collision Grid Map

In order to generate the Collision Grid Map, the map is set 150 * 100 in size and 0.01 m in resolution. Based on the sensor coordinate, x, y with the width of 1.5 m, the length of 1.0 m and depth data are used up to 2 m. When data is measured in the collision situation, Figure 5 shows the result of the Collision Grid Map. The example image was captured by a depth camera sensor shown in Figure 6a. Figure 6b shows raw 3D point cloud data. Figure 6c,d show the result of Collision Grid Map that consists of the first channel and second channel. In the second channel about depth data, the result shows that the nose of the cow becomes green due to the nearest distance based on a sensor coordinate. Similarly, Figure 7 shows results of the Collision Grid Map about a non-collision situation. Figure 7a,b show the image and 3D point cloud data each. Figure 7c,d show the result of the Collision Grid Map.

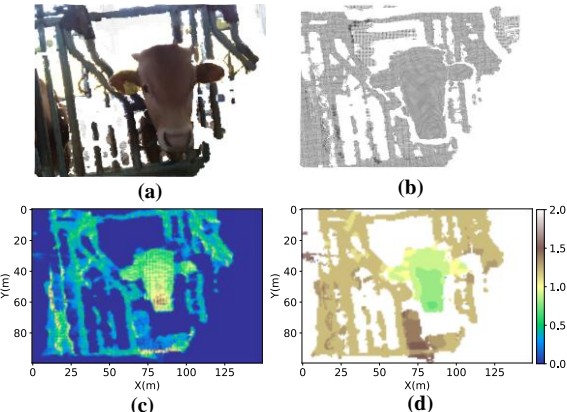

**Figure 6.** Result of the Collision Grid Map about the collision situation. (**a**) The example image that was captured by a depth camera sensor. (**b**) Raw 3d point cloud data. (**c**) The result of CGM in first channel. (**d**) The result of CGM in second channel.

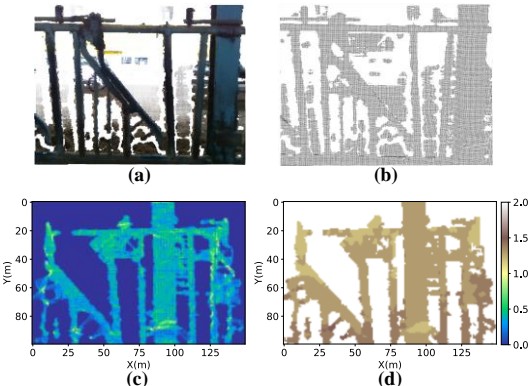

**Figure 7.** Result of the Collision Grid Map about the non-collision situation. (**a**) The example image that was captured by a depth camera sensor. (**b**) Raw 3d point cloud data. (**c**) The result of CGM in first channel. (**d**) The result of CGM in second channel.

Table 1 shows a comparison of data size by types. The data format unifies pickle to compare data sizes. Raw point cloud data size averages 4508.9 KB per data and the Collision Grid Map data size is 5.7 KB per data. The Collision Grid Map data size shows a significant reduction relative to raw point cloud data due to a transformation more than 100,000 raw data to $150 \times 100$ size of pixel data. Vox grid, which is set 200 * 200 * 200 size, 0.01 m resolution and hit map type for VoxNet [25] data size shows 8.0 KB. The size of voxnet data is relatively larger than the Collision Grid Map data size since map size is different.

**Table 1.** Comparison of data size by types.

| Data | Size (KB) |
|---|---|
| Raw point cloud | 4508.9 |
| Vox grid data | 8.0 |
| Collision Grid Map | 5.7 |

*3.3. Convolutional Neural Network Algorithm Using the Collision Grid Map*

A non-collision situation is defined when there is a fence only without any cow and in the case of the collision situation when cows stick their head out of the fence. A Convolutional Neural Network architecture using a Collision Grid Map consists of a convolutional layer, a pooling layer, and a fully connected layer. Figure 8 shows a block diagram of the network architecture. The convolutional layer is 3 * 3 in size and uses a Rectified Linear Unit (ReLu) activation function. The pooling layer is 2 * 2 in size and uses a *max* function. A dully connected layer has 1024 neurons. Then the Adam optimizer [29] is used for optimization.

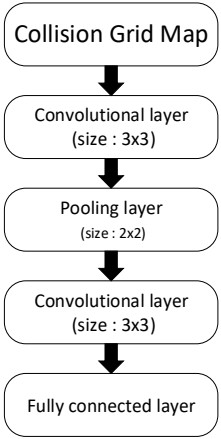

**Figure 8.** Block diagram of the network architecture.

Figure 9 shows visualization of the convolutional layer that is learned as a Collision Grid Map. The first convolutional layer is 3 * 3 in size and has 20 filters. In addition, the second convolutional layer is 3 * 3 in size and has 20 * 40 filters.

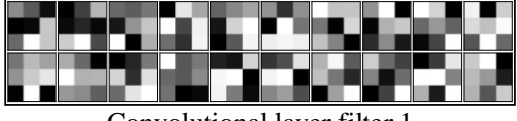

Convolutional layer filter 1

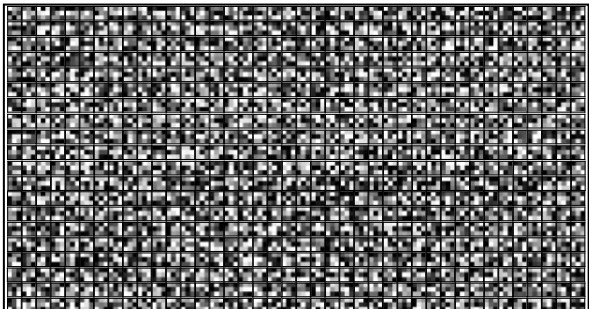

Convolutional layer filter 2

**Figure 9.** Visualization of the convolutional layer that is a learned Collision Grid Map.

The learned filter is able to detect the local features, when the Collision Grid Map is put in the neural network. The appearance of the filtered Collision Grid Map is shown in Figures 10 and 11. Figure 10 is an example of the collision situation. Similarly, Figure 11 is an example of the non-collision situation. The Collision Grid Map that is filtered by the first convolutional layer is shown in a filtered Collision Grid Map1. In addition, the Collision Grid Map that is filtered by a second convolutional layer is shown in a filtered Collision Grid Map2. The first convolutional layer is applied to the Collision Grid Map and the reacted edges are extracted from filtered collision grid map1. In the second convolutional layer, the network reacts as the distance, according to depth data as well as edges.

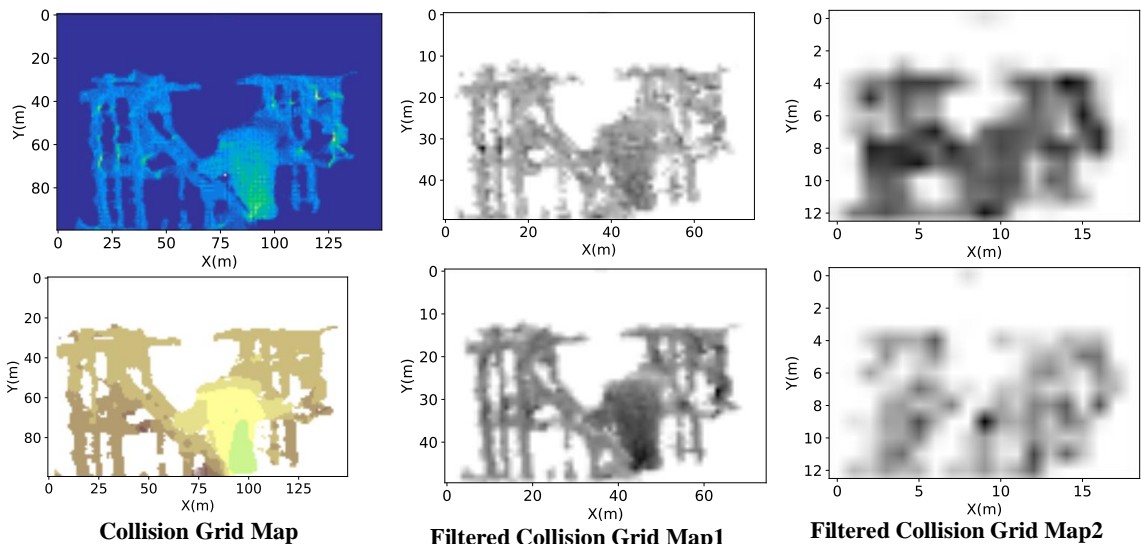

**Collision Grid Map**　　　　**Filtered Collision Grid Map1**　　　　**Filtered Collision Grid Map2**

**Figure 10.** Appearance of the Filtered Collision Grid Map about the collision situation.

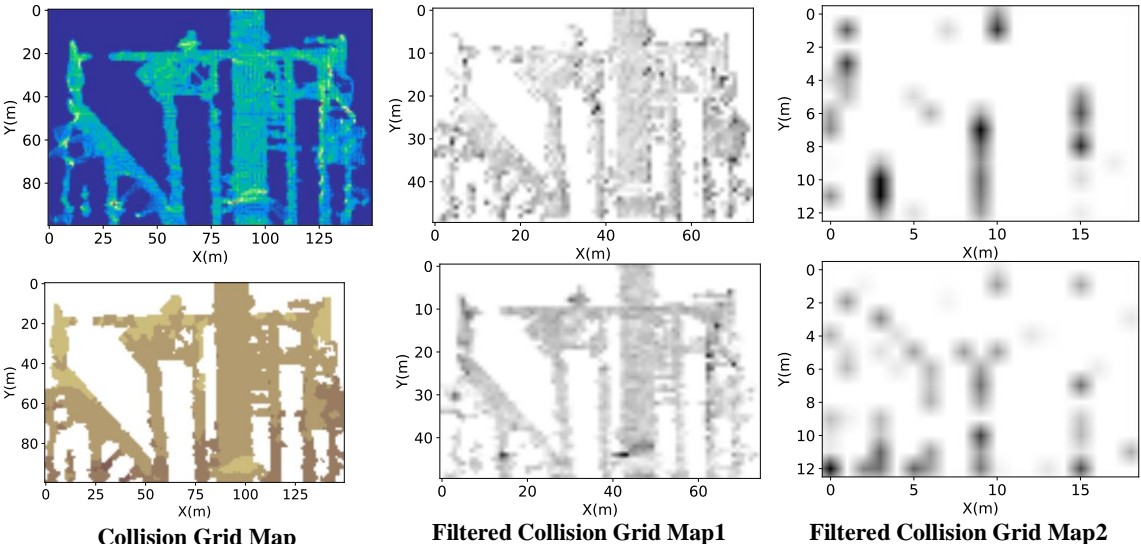

**Collision Grid Map**　　　　**Filtered Collision Grid Map1**　　　　**Filtered Collision Grid Map2**

**Figure 11.** Appearance of the Filtered Collision Grid Map about the non-collision situation.

The graph for comparison of accuracy for VoxNet and Convolutional Neural Network using Collision Grid Map is shown in Figure 12 and Table 2. The accuracy is measured with a test set of collision and non-collision situations. Accuracy for Convolutional Neural Network using the Collision Grid Map is measured 19.2% higher than VoxNet.

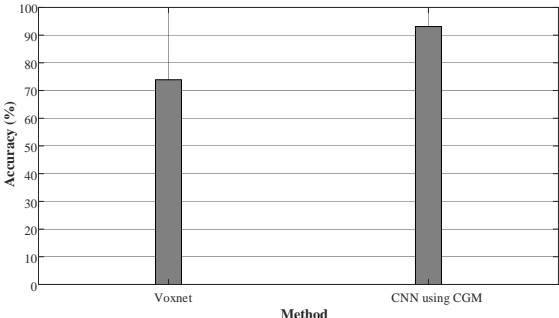

**Figure 12.** Comparison of accuracy for VoxNet and the Convolutional Neural Network using the Collision Grid Map.

**Table 2.** Comparison of accuracy results shown in Figure 11.

| Method | Accuracy (%) |
|---|---|
| Vox grid data | 73.9 |
| Collision Grid Map | 93.1 |

Figure 13 shows the accuracy of detecting each situation according to the method. The accuracy of each situation is the failure that the end effector may collide with a cow when the cow sticks its head out and collides with the robot. When the cow actually sticks its head out, the Convolutional Neural Network using the Collision Grid Map is measured 92.7% whether or not the robot can classify the situation that the cow really sticks its head out. Voxnet is measured 5.4% less accurately than the Convolutional Neural Network using a Collision Grid Map method.

| CNN using Collision Grid Map | Collision situation | Non Collision situation |
|---|---|---|
| Collision detection | 92.7% | 6.5% |
| Non Collision detection | 7.3% | 93.5% |

| Voxnet | Collision situation | Non Collision situation |
|---|---|---|
| Collision detection | 87.3% | 42.2% |
| Non Collision detection | 12.7% | 57.8% |

**Figure 13.** Accuracy of the detection of each situation according to the method.

## 4. Conclusions

In this study, the Collision Grid Map scheme is proposed from point cloud data to predict collision between the end effector and the cow. Generated Collision Grid Map is applied to a Convolutional Neural Network. The Collision Grid Map is converted from 3D point cloud based on a probability scheme and considered an invariant permutation problem when a 3D point cloud is applied to a Convolutional Neural Network. Furthermore, the convolutional layer is learned as the safety distance in the collision situation as well as local features from the x-y plane when the Collision Grid Map is used. We show that the Collision Grid Map can be effectively managed with an aspect to the size of data. The result of the field tests and experiments performed in this study shows that the Convolutional Neural Network using the Collision Grid Map method can obtain robust performances and better prediction in a collision situation. The Collision Grid Map using point cloud data can be applied to 3D lidar, solid state lidar, and a 3D depth camera.

In the future, we would like to investigate the detection of the position method into our proposed scheme. Then, it is expected to be used for path planning of the manipulator to prevent collision. This is able to reduce the duration of time when the mobile robot feed the cows.

**Author Contributions:** Conceptualization, J.H.J. and C.-B.M.; methodology, J.H.J.; validation J.H.J. and C.-B.M.; writing—original draft preparation, J.H.J.; writing—review and editing, C.-B.M.; visualization, J.H.J.; All authors have read and agreed to the published version of the manuscript.

**Funding:** This work was carried out with the support of "Cooperative Research Program for Agriculture Science and Technology Development (Project No. PJ01386005)" Rural Development Administration, Republic of Korea.

**Conflicts of Interest:** The authors declare no conflict of interest.

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
