# Peer review of "Learning Collision Situation to Convolutional Neural Network Using Collision Grid Map Based on Probability Scheme"

_applsci, doi:10.3390/app10020617_

Round 1
Reviewer 1 Report
This study proposes a Collision Grid Map(CGM) using 3d point cloud data to
predict the collision between the cattle and the end effector of a manipulator in the barn environment via CNN.
The experimental section should include details on the barn used for the present study.
There are too many grammar errors and the paper can not read properly. It requires extensive re-writing.
Reviewer 2 Report
The paper can be accepted after a minor review. Authors are invited to follow some recommendations in the revised version of the paper before its publication:
- The scheme of the paper sections inserted in the Introduction should be adjusted properly as it does not coincide with the numeration:
Sect. 2, the Collision Grid Map scheme and Convolutional Neural Network algorithm using Collision Grid Map
Sect. 3,The experimental setup and results
Sect. 4, The concluding remarks
- Data contained in table 1 could be explained with comments more in detail.
- The Conclusion could be better formulated as it does not tell something else from the abstract figuring quite the same. Instead it would be more appropriate to read in the conclusion more precise information on results and further application of the method in the field or other fields.
- As the language is concerned, it is not very fluent sometimes but there are not too many relevant errors in the paper. Please check and correct the following sentences of the sections below:
Abstract
- The method is proposed that Collision Grid Map scheme is composed of 2-channel.
Introduction
- Many studies[1]-[4] are proposed that robot is suitable for feeding and monitoring livestock in barn environment.
- Recently, it was proposed Convolutional Neural Network(CNN) algorithm[9]-[12] based on image data for object classification.
- ResNet[12] addresses the degradation problem using for residual learning method. Unlike images that is regularized pixel, 3D point cloud features unordered data.
- PointNet[17] is the network which is directly learned by point cloud segmented the features of 3d objects and demonstrated high performance in classifications.
- Each of cell in first channel is computed by distribution of location data in x-y plane
- This method is not transformed locational feature of point cloud in x-y plane data.
Section 2
- log odds notation can be calculated using present point cloud data and previous point cloud data. when initial grid map is free state, initial Collision Grid Map can be omitted.
- Value of cell in first channel can be seen probability about location in x-y data.
- However, there is required big data, due to the different appearance despite of same object.
- Network is learned spatial feature about collision and non-collision situation in convolutional layer
- When filter size of convolutional layers is w in N*N size, equation (7) is shown the learned convolutional layer for generated Collision Grid Map.
Section 3
- network reacts not only edge but also distance according to depth data
- Figure 9 and 10 are the collision situation and the non-collision situation each.
Conclusion
- And it is transformed x-y plane data and depth data to grid map using probability method in 3d point cloud data.
- And Convolutional layer is learned the safety distance in the collision situation as well as local features from x-y plane when Collision Grid Map is used.
